# Combination of CENP-B Box Positive and Negative Synthetic Alpha Satellite Repeats Improves De Novo Human Artificial Chromosome Formation

**DOI:** 10.3390/cells11091378

**Published:** 2022-04-19

**Authors:** Koei Okazaki, Megumi Nakano, Jun-ichirou Ohzeki, Koichiro Otake, Kazuto Kugou, Vladimir Larionov, William C. Earnshaw, Hiroshi Masumoto

**Affiliations:** 1Laboratory of Chromosome Engineering, Department of Frontier Research and Development, Kazusa DNA Research Institute, 2-6-7 Kazusa-Kamatari, Kisarazu 292-0818, Japan; mnakano@kazusagt.com (M.N.); johzeki@kazusa.or.jp (J.-i.O.); kotake@kazusa.or.jp (K.O.); kkugou@kazusa.or.jp (K.K.); 2Public Relations and Research Promotion Group, Kazusa DNA Research Institute, 2-6-7 Kazusa-Kamatari, Kisarazu 292-0818, Japan; 3Developmental Therapeutics Branch, National Cancer Institute, Bethesda, MD 20892, USA; larionov@mail.nih.gov; 4Wellcome Trust Centre for Cell Biology, Edinburgh EH9 3BF, UK; bill.earnshaw@ed.ac.uk

**Keywords:** HAC, chromatin, alphoid, CENP-B, histone, centromere, CENP-A, heterochromatin

## Abstract

Human artificial chromosomes (HACs) can be formed de novo by introducing large (>30 kb) centromeric sequences consisting of highly repeated 171-bp alpha satellite (alphoid) DNA into HT1080 cells. However, only a subset of transformed cells successfully establishes HACs. CENP-A chromatin and heterochromatin assemble on the HACs and play crucial roles in chromosome segregation. The CENP-B protein, which binds a 17-bp motif (CENP-B box) in the alphoid DNA, functions in the formation of alternative CENP-A chromatin or heterochromatin states. A balance in the coordinated assembly of these chromatin states on the introduced alphoid DNA is important for HAC formation. To obtain information about the relationship between chromatin architecture and de novo HAC formation efficiency, we tested combinations of two 60-kb synthetic alphoid sequences containing either tetO or lacO plus a functional or mutated CENP-B box combined with a multiple fusion protein tethering system. The combination of mutated and wild-type CENP-B box alphoid repeats significantly enhanced HAC formation. Both CENP-A and HP1α were enriched in the wild-type alphoid DNA, whereas H3K27me3 was enriched on the mutant alphoid array. The presence or absence of CENP-B binding resulted in differences in the assembly of CENP-A chromatin on alphoid arrays and the formation of H3K9me3 or H3K27me3 heterochromatin.

## 1. Introduction

The centromere is an essential chromosome region involved in regulating several pathways that control chromosome segregation during mitosis. On all normal human chromosomes, centromeres are formed at alpha satellite (alphoid) DNA, a highly variable array of highly repeated 171-bp monomer units that can be up to several megabases in size [1]. A histone H3 variant, CENP-A, defines the site of functional centromere formation epigenetically and is a key conserved factor among most eukaryotic centromeres [2,3,4]. CENP-A-containing chromatin assembles in a subdomain of the alphoid DNA at each human centromere [5]. CENP-A chromatin marks the interphase centromere complex (ICEN)/constitutive centromere-associated network (CCAN), where, during mitosis, more than 100 different proteins (hMis12 and CENP-C, -E, -F, -H, -I, and -K through -U, among others) are assembled to form a kinetochore structure that interacts with spindle microtubules to control chromosome movement and sister chromatid separation [6,7,8,9,10,11,12,13].

Although CENP-A chromatin covers a relatively short 40–160 kb area at neocentromeres, which assemble on non-alphoid DNA [14,15], it covers approximately 40% of X chromosome alphoid, ranging from 630 kb to 1.8 Mb, depending on the length of the alphoid array [16]. CENP-A assembles on either the D17Z1 or D17Z1-B array in the human chromosome 17 higher-order repeat (HOR), whereas heterochromatin exists on both HORs [17]. Thus, the majority of the megabase-order alphoid DNA array is assumed to be covered by non-CENP-A chromatin, principally heterochromatin. This is known as the pericentromere region. In the heterochromatin of the pericentromere, sister chromatid cohesion is maintained until the end of metaphase [18]. However, how these distinct chromatins are assembled and maintained in the huge homogeneous repeats of the alphoid DNA remains unclear.

The alphoid DNA of all normal human chromosomes except the Y chromosome has a 17-bp motif to which CENP-B binds, called the CENP-B box, which appears once every approximately two alphoid repeats (171-bp units) [19,20,21]. However, CENP-B gene knockout mice are viable [22,23,24]. The CENP-B box is also absent from human neocentromeres [25,26]. Interestingly, although CENP-B is not essential, it contributes to the fidelity of centromere function via interactions with CENP-A and CENP-C [27,28,29].

Similar to artificial chromosome formation in budding yeast [30] and fission yeast [31], alphoid DNA introduced into HT1080 human fibrosarcoma cells can efficiently form de novo human artificial chromosomes (HACs) [32,33]. HAC formation is accompanied by multimerization of the introduced alphoid BAC/YAC into megabase-sized arrays accompanied by rearrangement within the arrays [33,34,35,36,37,38,39]. The centromere and heterochromatin assemble on the concatemerized alphoid BAC/YAC DNAs [36,40,41,42] (Appendix A). CENP-B and the CENP-B box are required for de novo CENP-A assembly and HAC formation with introduced alphoid DNA [42,43,44].

CENP-B also induces heterochromatinization on alphoid DNA integrated into ectopic chromosomal arm regions [44]. In addition to the well-known CENP-B-interacting factors CENP-A and CENP-C, CENP-B recruits other chromatin proteins, including the open chromatin modifier ASH1L or heterochromatin factor HP1/Suv39h1 in a mutually exclusive manner [45]. Importantly, this reveals an unusual ability of CENP-B to create either an open or closed chromatin structure on transfected long alphoid DNA [44,45].

HACs can be formed not only with natural centromere-derived alphoid clones but also with synthetic alphoid DNA containing an artificial insertion of the tetO sequence (alphoid^tetO^) [46]. Experiments, in which tetR fusion proteins were tethered on a HAC consisting of alphoid^tetO^ repeats, revealed the importance of the balance between an open (for centromere) and closed (for heterochromatin) chromatin state of the alphoid DNA [45,46,47,48,49,50,51]. Thus, the alphoid^tetO^ HAC has contributed greatly to elucidating the epigenetic chromatin status required for centromere and HAC assembly de novo.

A drawback of using synthetic alphoid^tetO^ sequences for de novo HAC formation is that introduced alphoid^tetO^ BACs form HACs much less frequently (~5%) than natural alphoid BAC/YAC clones (~30%) [42,46]. In addition, tethering of a tetR fusion chromatin modifier on the HAC consisting of a single synthetic alphoid^tetO^ array uniformly changes chromatin throughout the whole HAC.

Therefore, in this paper, we constructed BACs carrying combinations of synthetic alphoid repeats containing lacO or tetO that permit tethering of multiple chromatin modifiers, plus combinations of wild-type or mutant CENP-B box synthetic alphoid monomers and used them for HAC formation assays. Chromatin analysis of the resulting HACs showed that the HAC formation frequency could be improved by allowing each of the two synthetic alphoid repeats to form a distinct chromatin region. The combinations of wild-type or mutant CENP-B box synthetic alphoid arrays improved the efficiency of HAC formation and caused differences not only in the assembly of CENP-A chromatin but also in the formation of H3K9me3 or H3K27me3 heterochromatin on alphoid arrays.

## 2. Materials and Methods

### 2.1. BAC Construction

The method to clone the tandemly chained synthetic alphoid repeats to 60 kb or 120 kb in a BAC vector was described previously [43].

### 2.2. Transfection

Cell culture and transfection of HT1080 cells were described previously [45,46]. FuGENE HD (Promega Corp., Madison, WI, USA; E2312) or Viafect (Promega, Beijing, China; E4981) was used as the transfection reagent. For transient protein tethering to the synthetic alphoid, 750 ng of the tethering plasmid pJETY3 [49], which can express a tetR-EYFP-fusion protein, and 750 ng of pJELC2 (this study), which can express a lacI-CLIP-fusion protein, were mixed with 100 ng of synthetic alphoid BAC and transfected into HT1080 cells grown in a 3.5-cm dish. One day after transfection, the cells were trypsinized and transferred into a 10-cm dish for further cultivation without selection with antibiotics. One week after transfection, selection with 400 μg/mL geneticin for a neo gene present on the BAC vector was started. For the HAC assay, 1 μg of the synthetic alphoid BAC was transfected into HT1080 cells grown in a 3.5-cm dish with or without 250 ng of each of the tethering plasmids. One day after transfection, the cells were trypsinized and transferred into a 10-cm dish with appropriate dilution for further cultivation with 400 μg/mL of geneticin.

### 2.3. FISH and ChIP

Preparation of chromosome spreads and subsequent fluorescence in situ hybridization (FISH) was described previously [46], except that we enriched for mitotic cells by treatment with 300 nM TN-16 (FUJIFILM Wako Pure Chemical Corp., Doshomachi Osaka, Japan) for 2 h. An evaluation of HAC formation was performed, as described previously [33].

The ChIP and subsequent qPCR and competitive PCR were described previously [45]. Anti-HP1α was purchased from abcam (Cambridge, UK; ab77256). PCR primers were from the literature [40], except N11F5 (5′-GGGATCACTAGCAATAAAAGGTAGAC-3′) and N11R6 (5′-TCCTTCTGTCTCGTTTTTATGGC-3′) used for competitive PCR of tetO vs. lacO, and N11F8 (5′-AGACAGAAGCATGCTCAGAAAC-3′) and N11R11 (5′-CTACCTTTTATTGCTAGTGATCCC-3′) for CENP-B box wild-type vs. mutant.

## 3. Results

### 3.1. Multiple Tethering and Competitive PCR for Chromatin Analysis of HACs with Two Synthetic Alphoid Repeats

To investigate whether the formation of distinctive chromatin domains by tethering fusion proteins can improve the efficiency of HAC formation, we generated a new series of synthetic alphoid repeat arrays with lacO or tetO insertions. The synthetic alphoid repeats in the present study were designed based on an 1880-bp 11mer higher-order repeat (HOR) type I on chromosome 21 (21-I; Figure 1a and Appendix A and Ikeno et al. [52]). In our previous studies, we used a repeat of the synthetic alphoid dimer combining a chromosome 17 alphoid monomer unit and a consensus monomer unit [46] or a synthetic alphoid dimer from the 21-I array (Ohzeki et al. [49]).

CENP-B box-mutant versions for each alphoid containing lacO or tetO were also synthesized by replacing all CENP-B boxes with mutated sequences. These synthetic HORs (CENP-B box wild-type 11mer alphoid^lacO^:LB, CENP-B box wild-type 11mer alphoid^tetO^:TB, CENP-B box mutant 11mer alphoid^lacO^:Lm, CENP-B box mutant 11mer alphoid^tetO^:Tm) were designed to be amplified and distinguished with a common primer set by competitive PCR after chromatin immunoprecipitation (ChIP; Figure 1a and S2). Therefore, their ratios can be maintained and quantified exactly even after PCR amplification by distinguishing between the presence or absence of restriction enzyme sites: *Eco*RV for whether the CENP-B boxes are wild-type or mutant, and *Sac*I for tetO or lacO (Figure 2b,c).

These 1880-bp-unit synthetic HORs were tandemly ligated to 32 copies over 60 kb via a method involving five rounds of cleavage and ligation cycles using *Nhe*I and *Spe*I restriction enzymes, which recognize the ends of the 1880-bp synthetic HORs. Both restriction sites can be ligated but are no longer recognized by either enzyme after the ligation (Figure 1b,c). Alphoid repeats with a length of 60 kb, here named pBAC11.32LB, pBAC11.32Lm, pBAC11.32TB, and pBAC11.32Tm, contain long enough (>30 kb) stretches of continuous alphoid sequence to generate de novo HACs when introduced into human fibrosarcoma derived HT1080 cells [42]. Two of these four synthetic alphoid arrays were joined (120 kb) to obtain pBAC11.64LBTB and pBAC11.64LmTB (Figure 1b–d).

### 3.2. De Novo CENP-A Chromatin Formation Efficiency and Effectiveness of Fusion Protein Tethering on Synthetic Alphoid Arrays

We first examined the CENP-B-dependent de novo CENP-A chromatin-forming activity of this new series of synthetic alphoid arrays (Figure 2). The CENP-B box wild-type alphoid (LB or TB) and mutant alphoid (Lm or Tm) BACs were mixed 1:1 and introduced into HT1080 cells. ChIP and competitive PCR were carried out 2 weeks after DNA transfection (Figure 2a). The ChIP recovery with anti-CENP-A antibody was 0.4% for cells transfected with TB+Tm, 0.3% for cells transfected with TB+Lm, and 0.1% for cells transfected with LB+Lm or LB+Tm (Figure 2d). Competitive PCR also showed that the CENP-B box wild-type/mutant ratio was 10 for cells transfected with TB+Tm or 16 for cells transfected with TB+Lm. This ratio was decreased to 4–5 for cells transfected with LB+Lm or LB+Tm (Figure 2c,d). The ChIP recovery with anti-CENP-B antibodies was also 0.4% for cells transfected with TB+Tm or 0.3% for cells transfected with TB+Lm, but was 0.2% for cells transfected with LB+Lm or LB+Tm. Competitive PCR revealed that the CENP-B box wild-type/mutant ratio was > 32 for cells transfected with TB+Tm or TB+Lm, but it was 20 for cells transfected with LB+Lm or 28 in cells transfected with LB+Tm (Figure 2c,d). Therefore, both CENP-A and CENP-B preferentially assembled on the wild-type CENP-B box alphoid arrays, and tetO insertion into these synthetic alphoid arrays allowed a more efficient recovery of CENP-A or CENP-B assemblies than lacO insertion.

**Figure 2 cells-11-01378-f002:**
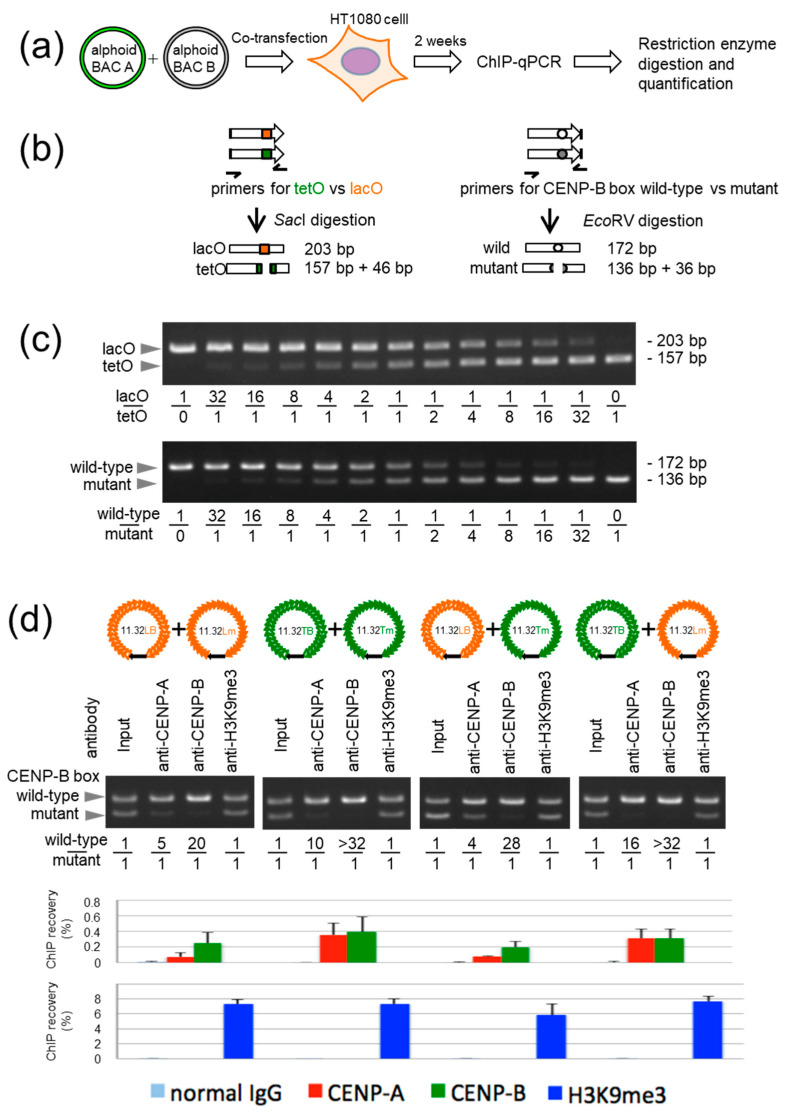
De novo CENP-A chromatin formation efficiency and effectiveness of fusion protein tethering on synthetic alphoid arrays. (**a**) Diagram of the transient ChIP assay to evaluate CENP-B-dependent CENP-A chromatin formation. (**b**) Diagram of the competitive PCR to quantify the ratio of the two types of alphoids within the immunoprecipitated chromatins. (**c**) Reference gel used to quantify the ratio of the two types after 35 cycles of PCR. (**d**) Top, combinations of the four types of synthetic alphoid arrays introduced into HT1080 cells and subjected to ChIP analysis 2 weeks after transfection. Middle, results of the competitive PCR. Bottom, results of qPCR. For the competitive PCR, the intensities of the 172-bp bands for the CENP-B box wild-type alphoids and 136-bp bands for the mutant alphoids were quantified according to the reference gel in (**c**). Error bars indicate standard deviation (*n* = 3).

In contrast, the ChIP recovery with anti-H3K9me3 2 weeks after DNA transfection was much higher (6–8%), and the wild-type/mutant ratio remained at 1 in competitive PCR (Figure 2c,d). Therefore, differences between lacO and tetO or between the wild-type and mutant forms of the CENP-B box did not cause large differences in H3K9me3 content within the initial 2 weeks. The results also indicated that it is reasonable to assign TB— which is naturally more active for CENP-A assembly—to the place inducing CENP-A chromatin assembly by fusion protein tethering, and LB or Lm—which are less active for CENP-A assembly—to the place inducing heterochromatin assembly.

Next, we examined the tethering specificity of the synthetic alphoids (Appendix A). The BACs Lm and Tm were mixed 1:1 and introduced into HT1080 cells simultaneously with plasmids expressing lacI-CLIP or tetR-EYFP, to confirm whether lacI-CLIP and tetR-EYFP can be tethered together and individually on the introduced alphoid DNAs. After 2 days of transfection, immunoprecipitation with anti-lacI or anti-GFP, followed by competitive PCR, detected specific binding of lacI to lacO and tetR to tetO, indicating that these synthetic alphoids were useful for dual tethering experiments.

### 3.3. Chromatin Manipulation by Multiple Fusion Protein Tethering

Two synthetic alphoid arrays, LB and TB, were combined into a BAC plasmid to generate pBAC11.64LBTB and then tethered with different chromatin-inducing factors to simultaneously induce separate chromatin structures in HT1080 cells. From previous studies [45,53], HJURP and HP1α are the strongest factors for assembly and elimination of CENP-A from the HAC centromere, respectively, when these fusion proteins are tethered to the alphoid^tetO^ HAC. Therefore, pBAC11.64LBTB was co-transfected into HT1080 with plasmids expressing tetR-EYFP-HJURP and lacI-CLIP-HP1α. After 4 days and 2 weeks, ChIP was carried out to test the chromatin assembly by the tethering effects (Figure 3a).

Individual alphoid ChIP recovery (Figure 3c) was calculated separately for TB and LB from the ratios obtained by competitive PCR (Figure 3b). Using non-immune IgG as a control, the background level of ChIP recovery of total alphoid DNA was < 0.03%. In another control, four days after transfection, ChIP performed with anti-CENP-A and with anti-HP1α antibodies following tethering of tetR-EYFP-alone and lacI-CLIP-alone, and again showed a background level of recovery for both TB and LB. Thus, centromere and heterochromatin induction on the transfected DNA by the binding of endogenous CENP-B appears to be below the detection level in this analysis.

Importantly, 4 days is the time when CENP-B-dependent CENP-A assembly begins to be detectable in the sensitive competitive PCR, regardless of tetO insertion [44,45]. In the competitive PCR experiments of Figure 3b, relative CENP-A assembly was detectable as biased to TB (lacO/tetO = 1/4) at this time point. At 4 days, 0.8% of TB was recovered by ChIP with anti-CENP-A in the sample with tetR-EYFP-HJURP tethering, whereas the recovery of LB was at a low background level (lacO/tetO = 1/31). This confirmed that specific CENP-A chromatin assembly is induced by HJURP tethering to TB. In the case of lacI-CLIP-HP1α tethering, 0.9% of LB was recovered by ChIP with anti-HP1α relative to a recovery of TB of 0.4%. Thus, tethering-dependent HP1α assembly on LB occurred but was less efficient (lacO/tetO = 2/1) than the tethering-dependent CENP-A assembly on TB (lacO/tetO = 1/31). The superior tethering specificity of tetR-EYFP relative to lacI-CLIP was also seen in the competitive PCR experiment in Appendix A. In simultaneous tethering of tetR-EYFP-HJURP and lacI-CLIP-HP1α, 0.2% of TB was recovered with anti-CENP-A, while the recovery of LB was background level (lacO/tetO < 1/32). In contrast, 0.8% of LB was recovered with anti-HP1α relative to a recovery of TB of 0.4% (lacO/tetO = 2/1).

Interestingly, in the tethering of tetR-EYFP-HJURP to TB, 0.4% of both TB and LB was recovered by anti-HP1α (lacO/tetO = 1/1). As a positive correlation between CENP-A chromatin and heterochromatin has also been suggested [51] (see Section 4.), the formation of CENP-A chromatin may induce HP1α assembly weakly on the alphoid DNA.

Two weeks after transfection, anti-CENP-A ChIP recovery of TB and LB in the control sample using tetR-EYFP-alone and lacI-CLIP-alone for tethering, increased to 0.6% and 0.15%, respectively (lacO/tetO = 1/4), indicative of some endogenous CENP-B-dependent CENP-A chromatin assembly on TB and LB. The higher recovery for TB/tetR than for LB/lacI is also consistent with the results in Figure 2. ChIP with anti-HP1α recovered 1.1% of TB and 1.2% of LB, indicating the progression of intrinsic heterochromatin formation on TB and LB (lacO/tetO = 1/1).

The tethering effect of tetR-EYFP-HJURP on CENP-A chromatin formation was slightly weakened in 2 weeks compared to 4 days, but 1.8% of TB (3-times greater than tetR-EYFP-alone) with single tethering of tetR-EYFP-HJURP (lacO/tetO = 1/9) and 3.0% of TB (5-times greater than tetR-EYFP-alone) with combination tethering of tetR-EYFP-HJURP and lacI-CLIP-HP1α was recovered with anti-CENP-A (lacO/tetO = 1/13). This effect was specific to TB. In contrast, the anti-HP1α ChIP recovery did not change significantly between LB and TB or between LB with tethering of lacI-CLIP-alone and lacI-CLIP-HP1α. We conclude that the tethering effect of tetR-EYFP-HJURP on CENP-A chromatin formation persists for at least the initial 2 weeks, whereas the tethering effect of lacI-CLIP-HP1α on the formation of HP1α-containing heterochromatin is no longer apparent after 2 weeks. Interestingly, tethering with the combination of lacI-CLIP-HP1α and tetR-EYFP-HJURP may still have a positive synergistic effect on CENP-A assembly.

The increased recruitment of HP1α by lacI-CLIP-HP1α may decrease after 2 weeks, either because tetR-EYFP-HJURP and lacI-CLIP-HP1α are expressed transiently and/or because of ongoing heterochromatin formation driven by endogenous CENP-B. ChIP recovery with anti-H3K9me3 was 9% for TB and 8% for LB in the tethering samples of tetR-EYFP-alone and lacI-CLIP-alone after 4 days, and slightly increased to 10% for TB and 12% for LB after 2 weeks. However, we did not observe a significant difference in H3K9me3 levels after tethering either the tetR-EYFP-alone and lacI-CLIP-alone controls or tetR-EYFP-HJURP, lacI-CLIP-HP1α after 2 weeks.

In summary, induction of either centromeric CENP-A chromatin or HP1α heterochromatin by combination tethering was effective in the early stages following the synthetic alphoid DNA introduction into HT1080 cells.

### 3.4. Efficiency of De Novo HAC Formation Using Two Distinct Alphoid Arrays

Temporary tethering with tetR-EYFP-HJURP and lacI-CLIP-HP1α induced parallel centromeric chromatin and heterochromatin assembly on TB and LB in the early stages of synthetic alphoid DNA introduction. Therefore, we tested whether such chromatin induction increases the efficiency of HAC formation by evaluating the HAC formation rates of newly obtained synthetic alphoid BACs. For the HAC formation assay, G418-resistant clones (cell lines) were isolated from BAC-introduced HT1080 cells, and chromosome spreads were analyzed by FISH with BAC DNA and chromosome 21 alphoid probes. Based on Ikeno et al. [33], we scored cell lines as carrying HACs if > 50% of the mitotic cells in each cell line had a HAC signal. Representative FISH images of the HAC cell lines obtained in this study are shown in Figure 4 and Table 1.

Furthermore, pBAC11.32LB and pBAC11.32TB were found to have the same HAC formation efficiency (4–5% of cell lines) as previous dimer-based synthetic alphoid^tetO^ BACs [46] (Table 1, Experiment A). However, this value is much lower than that of the wild-type 21-I 11mer-based synthetic alphoid repeats (28% on average) [42,43,46]. The HAC formation efficiency of pBAC11.64LBTB (120 kb) docked with pBAC11.32LB and pBAC11.32TB was also 4%, similar to the parent 60 kb BACs. When lacI-CLIP-HP1α and tetR-EYFP-HJURP were tethered to pBAC11.64LBTB, the HAC formation efficiency rose to 10% (Table 1, Experiment B). Although this change in HAC formation efficiency did not reach significance (χ^2^ = 1.2, *p* = 0.3), temporal chromatin manipulation, in the early stages of HAC formation, is expected to be one of the methods of improving de novo HAC formation.

Following these results, we performed the same multiple tethering in pBAC11.64LmTB, in which all CENP-B boxes in the LB array of pBAC11.64LBTB are mutated. The reason for testing this was that the combination of Lm and TB showed the most biased intrinsic CENP-A assembly activity as the result of competitive PCR in the early stages after BAC DNA transfection (Figure 2d). We expected that such intrinsic activity, combined with the effect of multiple tethering, might further improve the HAC formation rate. Surprisingly, pBAC11.64LmTB had a HAC formation efficiency of 24% without any tethering (Table 1, Experiment C). This is close to the HAC formation efficiency of BAC containing the natural alphoid repeats or wild-type 21-I 11mer-based synthetic alphoid (~30%), and is significantly higher than the value of pBAC11.32TB (2%) used as a control in this experiment (χ^2^ = 11, *p* = 0.001). It is also significantly higher (χ^2^ = 7, *p* = 0.01) than the data for pBAC11.64LBTB in Experiment B (4%). Even more surprisingly, when lacI-CLIP-HP1α and tetR-EYFP-HJURP were tethered to pBAC11.64 LmTB, the HAC formation efficiency decreased to 6% (χ^2^ = 4, *p* = 0.04).

Although it was unexpected, these results suggest that combining two different synthetic alphoid repeats, Lm and TB, to bias the individual chromatin assembly capacity as a DNA sequence, can improve the HAC formation efficiency of the synthetic alphoid DNA. Either multiple tethering of chromatin-inducing factors or the presence of combination arrays with one having and the other lacking CENP-B boxes could increase the HAC formation capacity, though the latter was more effective than the former. However, simultaneous use of the two systems to bias the individual chromatin assembly may be counterproductive in some cases.

### 3.5. Chromatin Structure of HACs Consisting of a Combination of CENP-B Box Wild-Type and Mutant Alphoid Arrays

Examining the chromatin structure of the HACs formed from pBAC11.64LmTB could potentially provide information on the optimal chromatin arrangement required for HAC formation and maintenance. Therefore, ChIP was performed on four HAC strains:C32-2 HAC cell line, obtained from pBAC11.64LBTB with lacI-CLIP-HP1α and tetR-EYFP-HJURP tethering;K9-3 and K11-3 HAC cell lines, obtained from pBAC11.64LmTB without tethering;H26-5 HAC cell line, obtained from pBAC11.64LmTB with lacI-CLIP-HP1α and tetR-EYFP-HJURP tethering (Figure 5, Table 1).

The recovery of synthetic alphoid arrays was split into Lm and TB (or LB and TB in the case of the C32-2 cell line) by calculation using the tetO/lacO ratios obtained by competitive PCR after ChIP analysis (Appendix A). During the multimerization of the introduced BACs, all samples nearly maintained the original tetO/lacO ratio of 1/1 (0.8~1.3).

The ChIP recovery of TB with both anti-CENP-A and anti-CENP-B antibodies was similar to that of chromosome 21 centromeric alphoid 11mer (21-I) for all HACs (0.4–0.8% for CENP-A, 0.2–0.6% for CENP-B), as expected from the initial assembly activity of CENP-A towards TB (Figure 2d and Figure 3b).

The recovery of Lm with anti-CENP-A or anti-CENP-B was much lower (less than one-tenth) than that of TB (Figure 5). We do not know why the recovery of TB by anti-CENP-B antibodies is higher in the H26-5 cell line than in the other cell lines. As the recovery of the chromosome 21 centromeric alphoid is also high in the same sample, it may be due to clonal variation of CENP-B expression in the cell line rather than structural differences in the HAC.

H3K4me2 is a histone modification for the open chromatin state that is also involved in CENP-A assembly [54]. ChIP recovery of TB with anti-H3K4me2 antibody was slightly higher in all HACs than the recovery of Lm or chromosome 21 centromeric alphoid. In contrast, ChIP with anti-H3K36me3 antibody showed a significantly higher recovery of TB than Lm in the LmTB-HACs in the K9-3, K11-3, and H26-5 cell lines. This is in good agreement with a recent report [45] that CENP-B enhances the CENP-A assembly on alphoid arrays by recruiting the H3K36methylase ASH1L.

The distribution of H3K27me3 modification on Lm, which is mutually exclusive with the H3K36me3 modification, was the exact opposite of the distribution of H3K36me3 on TB. The recovery of Lm with anti-H3K27me3 antibody was as high as 30–40% of input DNA, indicating that the inability of CENP-B to bind can explain the high levels of H3K27me3 facultative heterochromatin assembly on Lm. On the other hand, the distribution of H3K9me3, a marker of constitutive heterochromatin, was almost unbiased between TB and Lm (LB) in all HACs. In ChIP with anti-HP1α antibodies, the recovery of TB in all HAC cell lines was similar to that of centromeric alphoid on chromosome 21. The recovery of LB in the C32-3 cell line obtained by the tethering of HP1α to LB was similar to the recovery of TB, but the recovery of Lm in the K9-3 or K11-3 cell line was lower than the recovery of TB. These results are reasonable because HP1α assembly into the alphoid array is also affected by CENP-B binding [45]. Notably, even the H26-5 HAC cell line obtained by tethering HP1α to Lm had a lower recovery of Lm with anti-HP1α.

These results show that the chromatin structures of the obtained LmTB-HACs (K9-3, K11-3 and H26-5) are very similar to each other, regardless of the effect of the protein tethering aimed at intentional chromatin formation. This finding suggests the intrinsic cellular ability to assemble chromatin, due to the properties of the input alphoid DNA arrays. can increase the efficiency of HAC formation, and the unique sequence arrangement of the combination of Lm and TB (K9-3 or K11-3 cell line) is the most efficient of the configurations tested here.

## 4. Discussion

The HAC consisting of synthetic alphoid^tetO^ DNA has been highly useful in characterizing the effects of different chromatin conformations on centromere assembly and stability. For example, creating a bias towards heterochromatin or open chromatin by tethering chromatin modifiers on a whole alphoid^tetO^ HAC led to the disassembly or assembly of CENP-A, revealing that the balance between these two chromatin states is important [46,47,55]. However, when tetO or lacO sequences are inserted into the alphoid HOR (alphoid^tetO^ or alphoid^lacO^), the de novo formation efficiency for the HAC is much lower than that of the natural HOR-based alphoid arrays, and there is an urgent need to improve this efficiency.

Both CENP-A chromatin and heterochromatin assemblies, which serve as centromere/kinetochore and sister chromatid cohesion regions, respectively, are indispensable for chromosome segregation in mitosis. Therefore, manipulating chromatin modifications to optimize the balance between CENP-A chromatin and heterochromatin is expected to improve de novo HAC formation. However, tethering a tetR-fused chromatin modifier on the HAC consisting of a single homogeneous synthetic alphoid^tetO^ array changes the chromatin uniformly throughout the HAC. In the present study, we constructed BACs carrying new synthetic alphoid chimeras, LB and TB, and attempted to improve the HAC formation frequency by ensuring the formation of both CENP-A chromatin and heterochromatin separately via a multiple tethering system. The efficiency of HAC formation of pBAC11.64LBTB was 4%, but this increased to 10% when lacI-CLIP-HP1α and tetR-EYFP-HJURP were temporarily tethered to the synthetic alphoid arrays. This suggests that HAC formation can be improved by combining LB and TB and assigning them to heterochromatin and CENP-A chromatin, respectively, to ensure the formation of both chromatins.

Expecting that further chromatin specification of the synthetic two-alphoid chimera could further improve the de novo HAC formation efficiency, we performed similar experiments using a combination of Lm (CENP-B box mutant version of LB) and TB. However, pBAC11.64LmTB unexpectedly exhibited a much higher HAC formation efficiency of 24%, despite tetO and lacO insertions and without any fusion protein tethering. In contrast, when lacI-CLIP-HP1α and tetR-EYFP-HJURP were tethered to the synthetic alphoid of this combination of LmTB, the HAC formation rate decreased to 6%. Why did the repetition of LmTB, in which CENP-B boxes and mutant boxes appear alternately every 60 kb, significantly improve the HAC formation efficiency without any tethering compared to the repetition of LBTB, in which CENP-B boxes are distributed across the entire alphoid array? Why did the tethering to LmTB reduce HAC formation 4-fold, whereas the same tethering to LBTB enhanced it slightly? The only difference between LBTB and LmTB is the two bases in the multiple CENP-B boxes in LB and Lm, and whether CENP-B binds. Figure 6 shows our model of how de novo HAC formation differs between LBTB and LmTB or with and without tethering. We speculate that it is important for HAC establishment that a CENP-A chromatin core forms in only one (or very few) particular region(s) of the TB alphoid arrays, with heterochromatin forming on the remainder. Indeed, the CENP-A chromatin core of neocentromeres is often localized in a limited area of 40–160 kb [14,15]. Tethering HJURP to TB is effective in ensuring the formation of CENP-A chromatin, but it could potentially induce the formation of multiple CENP-A chromatin islands across the many TB alphoid arrays. Nevertheless, in the case of LBTB, the heterochromatin induced by tethering HP1α to the LB alphoid arrays invades adjacent TB arrays to reduce the number of CENP-A chromatin islands (Figure 6A). This model fits well with our results. The ChIP recovery of synthetic alphoids with anti-CENP-A antibody 2 weeks after tethering lacI-CLIP-HP1α and tetR-EYFP-HJURP to pBAC11.64LBTB was 3% (Figure 3) but eventually decreased to 0.8% in the established HAC C32-2 (Figure 5).

The HACs obtained using pBAC11.64LmTB without tethering formed naturally using endogenous chromatin proteins, depending on the presence or absence of the CENP-B boxes in the alphoid arrays. CENP-A, H3K36me3, and HP1α showed a biased distribution in CENP-B box-positive alphoid TB arrays in LmTB. In contrast, H3K27me3-rich facultative heterochromatin was formed in CENP-B box mutant alphoid Lm arrays. This pattern was reproduced in independent HAC clones (K9-3 and K11-3 cell lines), consistent with the property of CENP-B reported by Otake et al. [45] and consistent with what was previously seen when levels of H3K9me3 were lowered in the alphoid^tetO^ HAC [51]. Due to the high efficiency of HAC formation in LmTB, the chromatin arrangement and characteristics of HACs based on this LmTB array may represent a favorable configuration for HAC formation. Surprisingly, even if HP1α was tethered to the Lm arrays on the CENP-B box mutant side, the resulting HACs (H26-5 cell line) had a similar chromatin arrangement as HAC clones (K9-3 and K11-3 cell lines) obtained without tethering. In addition, HP1α was assembled more abundantly on the TB side rather than on the HP1α tethered Lm side. This suggests that HP1α is unstable on the CENP-B box mutant Lm alphoid arrays, and that its localization on the CENP-B box-positive alphoid arrays is important for HAC formation. In other words, without CENP-B binding, the induction of stable heterochromatin by HP1α tethering to the Lm arrays would be inhibited, leaving multiple CENP-A chromatin islands on the TB arrays induced by HJURP tethering. We suggest that this is the reason why the tethering to LmTB reduced the HAC formation frequency (Figure 6B,C).

Why is HAC formation more efficient with LmTB without tethering? CENP-B promotes the formation of CENP-A and/or H3K9me3 heterochromatin, however, on alphoid DNA immediately after DNA transfection, CENP-B also recruits ASH1L (H3K36 methylase) to create open chromatin conditions favorable to CENP-A chromatin formation. Over time, HP1/Suv39h1 heterochromatin formation mediated by CENP-B occurs [45]. Therefore, after functional CENP-A chromatin is established in any TB array on the HAC precursor, most of the remaining TB arrays switch from CENP-A to the heterochromatin side and maintain their integrity as a stable HAC with a single centromere.

A CENP-A chromatin maintenance mechanism mediated by kinetochore proteins may be involved in core centromere formation. In previous experiments using an alphoid^tetO^ HAC and ectopic alphoid^tetO^ site, many kinetochore components tethered to the alphoid^tetO^ arrays mediated the incorporation of new CENP-A into chromatin via a CENP-C and CENP-I-mediated pathway as a canonical mechanism of centromere assembly and maintenance [53,56,57,58,59]. As CENP-C and CENP-I also assemble in the region of HAC precursor alphoid DNA, where CENP-A chromatin and a functional kinetochore have been formed, CENP-B is assumed to prioritize the open chromatin formation and support the incorporation of CENP-A chromatin. In contrast, in TB arrays without this positive feedback loop, CENP-B would promote heterochromatinization to build a pericentromeric sister chromatid cohesion area. Overall, these actions would consolidate CENP-A chromatin in one or very few islands.

On the other hand, heterochromatin or HP1α assembled on the TB arrays in LmTB HACs may function, not only for sister chromatid cohesion but also for CENP-A chromatin maintenance. Martins et al. [51] found that tethering H3K9 demethylase JMJD2D to alphoid^tetO^ HAC to reduce H3K9me3 and HP1 levels, decreased CENP-A and CENP-C association in the long term, but had little effect on their levels in the short term. When JMJD2D tethering was dismissed from the HAC, the CENP-A chromatin in the HAC returned to its initial level. These observations indicate that H3K9me3 heterochromatin is also involved in the structural maintenance of CENP-A chromatin. If H3K9me3/HP1 needs to be localized close to the CENP-A chromatin for this mechanism, it may be one of the reasons for H3K9me3/HP1α localization on the TB arrays and not Lm.

Interestingly, Martins et al. [51] showed that reducing H3K9me3 levels by tethering JMJD2D, led to compensation for the heterochromatin function by increasing H3K27me3 levels instead. The high level of H3K27me3 assembly in the Lm arrays of the HAC in the present study may be explained by such a heterochromatin homeostasis function intrinsic to the cells. H3K36me-ase ASH1L is a trithorax group protein (Trx) and has been reported to be mutually exclusive with polycomb group (Pc)-mediated H3K27me3 facultative heterochromatin [60,61]. Another possible explanation is that H3K27me3 is excluded due to such mutually exclusive actions with H3K36me3 assembled on the TB arrays by the action of ASH1L recruited by CENP-B. Although the reason for the accumulation of H3K27me3 on the Lm arrays is not currently clear, it may function to insulate a particular TB island to form a core centromere while the other TB islands form H3K9me3/HP1α heterochromatin.

Though the effect in this study was relatively modest (approximately 2-fold), we attempted to improve the efficiency of HAC formation by manipulating chromatin assembly with a multiple tethering system. Surprisingly, the simple combination of synthetic alphoid arrays Lm (CENP-B box-negative) and TB (CENP-B box-positive) achieved a higher efficiency (approximately 5 to 6-fold) comparable to that of wild-type alphoid DNAs. It, therefore, proved to be more effective in HT1080 cells to rely on an intrinsic auto-assembly mechanism involving CENP-B for centromere/heterochromatin without tethering. However, such tethering can also break the epigenetic barrier to HAC formation in other strong heterochromatin-forming cell lines, such as HeLa [49]. By improving and optimizing the various conditions required for HAC formation, it should be possible to achieve 100% HAC formation efficiency for any cell type in the future.

## Figures and Tables

**Figure 1 cells-11-01378-f001:**
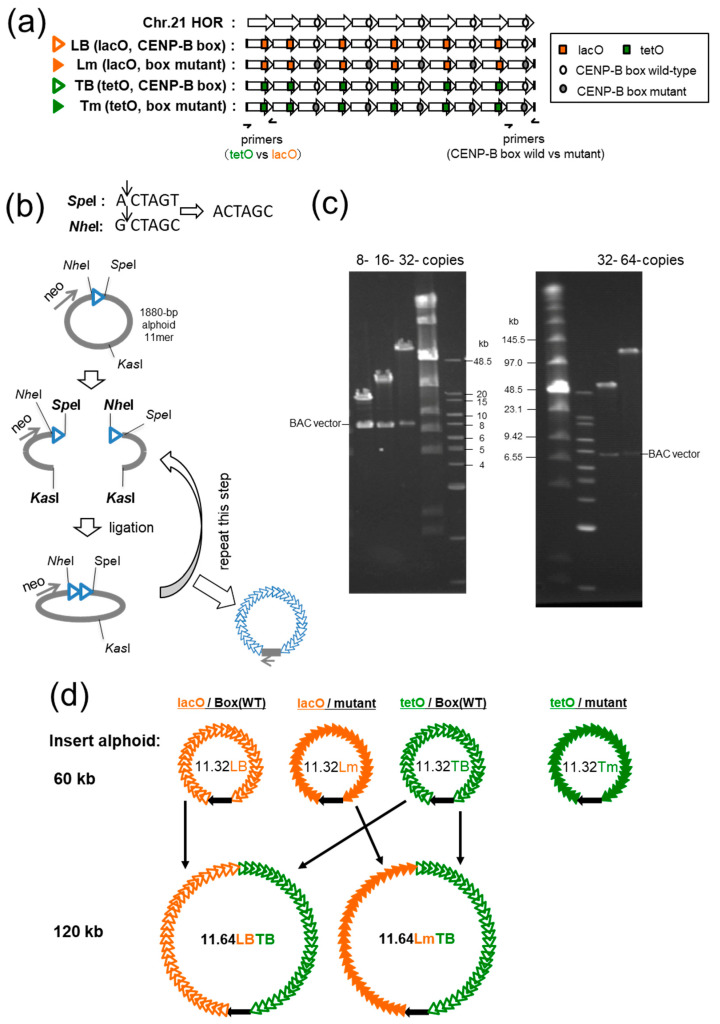
System for chromatin analysis of HACs with two synthetic alphoid repeats. (**a**) Synthetic alphoid design. Four synthetic alphoid sequences were designed based on 1880-bp higher-order repeat (HOR) type I on chromosome 21 (21-I; top) consisting of 11 alphoid repeats, 5 of them containing the CENP-B box sequence (open circle). LB, the lac operator (lacO) sequence (orange square) was introduced into 6 alphoid repeats at the positions corresponding to the CENP-B box. Lm, all 5 CENP-B boxes in LB were mutated (closed circle). TB, the tet operator (tetO) sequence (green square) was introduced instead of lacO. Tm, all 5 CENP-B boxes in TB were mutated. The positions of primers for the competitive PCR are indicated as half arrows. (**b**) Cartoon depicting extending the copy number of 1880-bp HOR units via cleavage and ligation cycles. (**c**) Pulsed-field gel electrophoretic images of the intermediate and final clones digested with *Bam*HI and *NotI* to release the insert from vectors. As an example, the left gel shows the TB clone series, and the right gel shows pBAC11.32LB and pBAC11.64LBTB. (**d**) Lineup of the synthetic alphoid BAC clones used in this study. Open and closed orange triangles represent LB and Lm repeats, respectively, and open and closed green triangles represent TB and Tm repeats, respectively. BAC vectors are shown as black arrows indicating the direction of neo gene transcription. Combination used to obtain BAC clones with a 120-kb insert are shown.

**Figure 3 cells-11-01378-f003:**
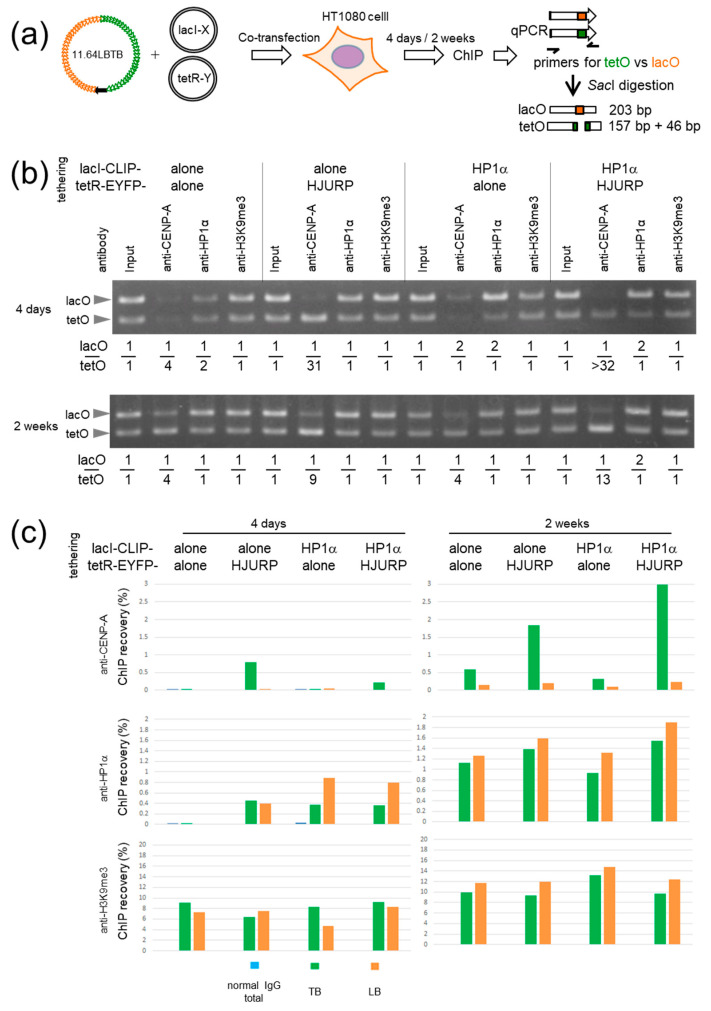
Tethering effect on chromatin formation in HAC precursors. (**a**) Cartoon of the experiment. (**b**) Results of competitive PCR after 4 days (upper) and 2 weeks (lower) of transfection. (**c**) Results of qPCR. Recovery of TB and LB was calculated from the lacO/tetO ratio obtained by the competitive PCR in (**b**) using the following formula; LB = (recovery × 2)/(ratio + 1), TB = (recovery × ratio × 2)/(ratio + 1).

**Figure 4 cells-11-01378-f004:**
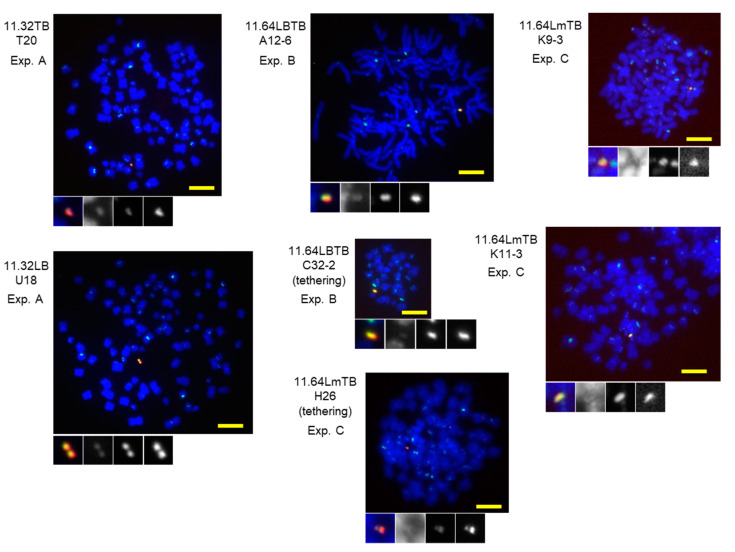
FISH analysis of the HACs obtained using the synthetic alphoid arrays. Chromosome spreads of the clones were hybridized with probes for α21-I (green) and BAC (red) and stained with DAPI (blue). Scale bar = 5 μm. Below each panel, the HAC area is magnified, and the color is split into black-and-white images for DAPI, α21-I and BAC, in that order.

**Figure 5 cells-11-01378-f005:**
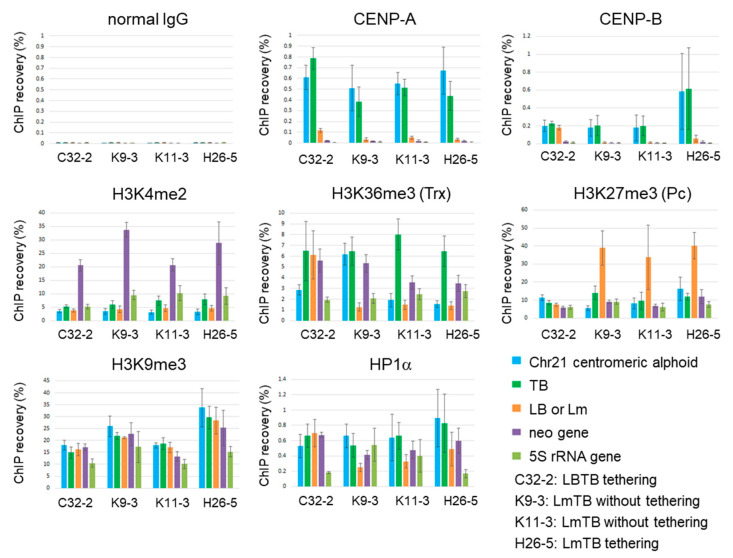
ChIP analysis of the HAC chromatin structure using synthetic alphoid arrays. Results of qPCR. Recovery of TB and LB was calculated from the lacO/tetO ratio obtained by the competitive PCR (Appendix A) as in Figure 3. Error bars indicate standard deviation (*n* = 3).

**Figure 6 cells-11-01378-f006:**
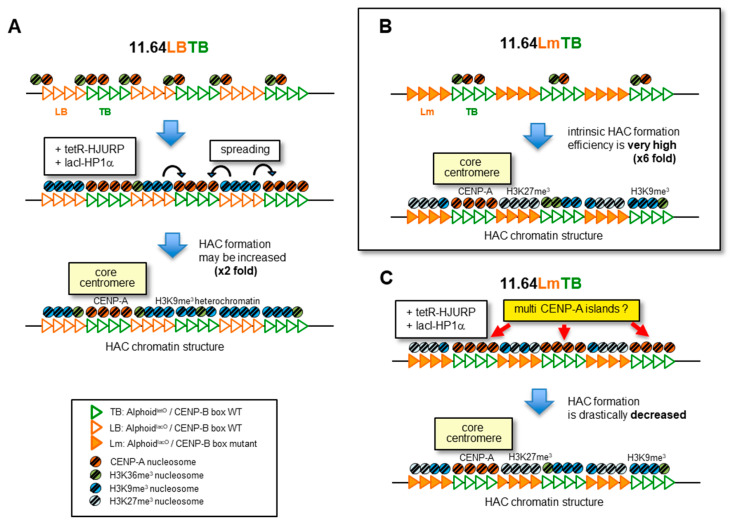
Model of chromatin assembly to establish HACs. (**A**) pBAC11.63LBTB with transient tethering of lacI-CLIP-HP1α and tetR-EYFP-HJURP. (**B**) pBAC11.63LmTB without tethering. (**C**) pBAC11.63LmTB with transient tethering of lacI-CLIP-HP1α and tetR-EYFP-HJURP. During the process of establishing HAC, the introduced alphoid BAC circular molecules are recombined and multimerized. Consequently, size variations of the alphoid DNAs arise, but for the sake of brevity, they are shown here at the same size [33,34,35,36,37,38,39].

**Table 1 cells-11-01378-t001:** De novo HAC formation efficiency by synthetic alphoid BACs.

Experiments	Alphoid BAC	Tethering	Analyzed Cell Lines	De Novo HAC Formation *	HAC Clone
Ohzeki et al., 2002 [43]	wild type 11.32	none	27	12 (44%)	
Okamoto et al., 2007 [42]	wild type 11.32	none	23	6 (26%)	
	wild type 11.64	none	42	7 (17%)	
Nakano et al., 2008 [46]	wild type 11.32	none	41	12 (29%)	
	BAC32-2mer(tetO)	none	46	2 (4%)	
A	pBAC11.32TB	none	19	1 (5%)	T20
	pBAC11.32LB	none	23	1 (4%)	U18
B	pBAC11.64LBTB	none	50	2 (4%)	A12-6
	pBAC11.64LBTB	tetR-EYFP-HJURP lacI-CLIP-HP1α	29	3 (10%)	C32-2
C	pBAC11.32TB	none	56	1 (2%) **	
	pBAC11.64LmTB	none	63	15 (24%) **	K9-3, K11-3
	pBAC11.64LmTB	tetR-EYFP-HJURP lacI-CLIP-HP1α	34	2 (6%) **	H26-5

* Number of cell lines that gave > 50% chromosome spreads contained a HAC. ** χ^2^ = 11, *p* = 0.001 between pBAC11.32TB and pBAC11.64LmTB. χ^2^ = 4, *p* = 0.04 between pBAC11.64LmTB with and without tethering.

## Data Availability

Not applicable.

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
