# Peer review of "Combination of CENP-B Box Positive and Negative Synthetic Alpha Satellite Repeats Improves De Novo Human Artificial Chromosome Formation"

_cells, 2022, doi:10.3390/cells11091378_

Round 1

Reviewer 1 Report

In the current research climate the advancements in disease diagnosis and management are being increasingly supported by new techniques such as CRISPR-Cas9. Using a de novo Human Artificial Chromosome (HAC) with either genes or repeated targeting a particular disease would revolutionize downstream prognosis of many current untreatable diseases. This also addresses the gap in optimization of such a process by using the previously established YAC or BACs which may have intended expression levels or worse, off site target effects. At present, de novo HACs may be formed by inserting satellite DNA repeats alphoid DNA into target cells but the efficiency is low. The authors address this issue by testing four different combinations of  BACs carrying synthetic alphoid sequences with tetO or lacO system to show that a combination of muted and WT CENP-B box increases the chances of HAC formation significantly.

Overall, I find this manuscript well written and well organized. The design of experiments is meticulously planned and the results section along with figures establishes the point. As CENPA and CENP-B proteins play a crucial role in centromere formation and position, this work can also be of interest to wider scientific community eg. Aedes aegypti which has a large chunk of genome in repeats and the sex locus is present in the centromere of chromosome 1. In addition, techniques allowing the manipulation of heterochromatin could benefit many genomes which possess a significant amount of repeats in their genome.

Minor comment-

In figure 4, I would like at least one inset where the FISH image is magnified, chromosome labelled and clearly showing the hybridization. In the current image, the resolution makes it harder to visualize the probes a-21-I and BACs.

Author Response

Thank you for reviewing our manuscript with high evaluation.

According your comment pointed by referee 1, we changed Figure 4, adding magnified HAC area split color into black-and-white images for DAPI, a21-I and BAC.

Reviewer 2 Report

In this manuscript, Okazaki and coworkers have compared the ability of large DNA sequences consisting of different modified alpha satellites to form Human artificial chromosomes (HACs) in HT1080 cells. They found that HAC formation is improved when both the mutated and the wild-type CENP-B boxes are included. The mechanisms are dissected in a clever way. Both CENP-A chromatin and heterochromatin are required for a normal centromere to function and binding of the CENP-B protein is involved in the control of those alternative chromatin states. The manuscript is very clearly written and the experiments are conclusive.

These findings are important for the people interested in the role of the cenpB protein in centromere function and may have interest for further developments in the field of HACs. Therefore, I strongly recommend publication of this manuscript in Cells.

Specific remarks:

-The term HOR appears for the first time on page 2 lane 7 and also in some figure legends but was never defined. Even if the term is pretty well known in the centromere community, I suggest to write the complete signification at least once.

-the last sentence of page 7 sounded not completely clear to me.

Author Response

Thank you for reviewing our manuscript with high evaluation.

According to your comments, we added the complete signification of HOR on page 2 lane 7, and inserted reference in the last sentence of page 7. The first sentence of the last paragraph of page 7 describing about Figure S3 was moved to the former paragraph.